# Simulating the next steps in badger control for bovine tuberculosis in England

**Graham C. Smith** *, Richard Budgey

National Wildlife Management Centre, Animal and Plant Health Agency, Sand Hutton, York, United Kingdom

* graham.smith@apha.gov.uk

## Abstract

Industry-led culling of badgers has occurred in England to reduce the incidence of bovine tuberculosis in cattle for a number of years. Badger vaccination is also possible, and a move away from culling was "highly desirable" in a recent report to the UK government. Here we used an established simulation model to examine badger control option in a post-cull environment in England. These options included no control, various intermittent culling, badger vaccination and use of a vaccine combined with fertility control. The initial simulated cull led to a dramatic reduction in the number of infected badgers present, which increased slowly if there was no further badger management. All three approaches led to a further reduction in the number of infected badgers, with little to choose between the strategies. We do note that of the management strategies only vaccination on its own leads to a recovery of the badger population, but also an increase in the number of badgers that need to be vaccinated. We conclude that vaccination post-cull, appears to be particularly effective, compared to vaccination when the host population is at carrying capacity.

## Introduction

Bovine tuberculosis, bTB, caused by *Mycobacterium bovis*, is a serious economic disease of cattle and in the British Isles its management in livestock is complicated by the involvement of the European badger (*Meles meles*), which may be responsible for about half of all cases in cattle [1]. Widespread and sustained badger culling has been shown to reduce bTB in cattle in England [2] during an experimental trial. This led to larger areas being licenced to undertake industry-led badger control by culling with 32 areas under control by the end of 2018 and analysis of the results indicating a significant reduction in bTB in cattle [3, 4] in those areas that had undertaken control for four years.

Intra-muscular vaccination of badgers with Bacillus Calmette-Guérin (BCG) has demonstrated reduced severity and disease progression [5], and indirect protection of cubs was observed in field studies [6]. Vaccination may therefore have a role to play in disease control in badgers although it is not yet certain what level of effect this would have on bTB incidence in cattle. Simulations comparing badger culling and vaccination have shown that quicker benefits are derived from culling [7–9], which is not unexpected since vaccination can only remove future infected animals.

**Data Availability Statement:** All relevant data are within the paper and its Supporting Information files.

**Funding:** This work was funded by Defra (Department for Environment and Rural Affairs) as

a request to investigate the topic. They played no role in study design, analysis or drafting the paper.

**Competing interests:** The authors have declared that no competing interests exist.

Now that a large portion of the High Risk Area for bTB is under licenced badger culling, a recent report suggested that "moving from lethal to non-lethal control of the disease in badgers is highly desirable" (i.e. vaccination) [10]. Here, we use an established simulation model [7, 11] to investigate potential exit strategies from the badger cull in England. We hypothesise that vaccinating badger post-cull will bring more immediate gains for vaccination compared to culling, than previous vaccination models which applied vaccination at carrying capacity. We therefore compare different culling options to maintain the reduced badger population size, badger trap and vaccination, and badger trap and vaccination plus fertility control. There is currently no oral vaccine available for badgers, so we assumed vaccination, and fertility control, was performed on trapped animals. This also ensures delivery to the same proportion of animals and similar effort for all strategies.

## Methods

### Model description

We used an established simulation model to predict the efficacy of different badger intervention methods with the objective of controlling levels of bTB (various continued proactive culling, vaccination or vaccination plus fertility control) in the badger population. The model, the rationale behind its design and development, and a sensitivity analysis are described elsewhere [7, 11, 12], with a full description of the model and parameter values in S1–S4 Appendices.

### Overview

We used the latest version of an individual based model which simulated the population dynamics of wild badgers, and the dynamics of bTB. The efficacy of a range of badger control methods to reduce bTB in cattle was investigated. The model's starting conditions were set to represent high incidence regions within England, where previous badger culling has taken place. By default, interventions occurred for five years across approximately 400 km$^2$ within a total area of 1600 km$^2$. Results were recorded for a range of measures inside in the intervention area, including the size of the badger population, and the levels of infection in badgers, after an initial four year badger cull to simulate an exit strategy from the licenced culling.

### Organisation of the model

The model was spatially explicit with badger territories organised by combining cells on a grid representing the landscape, and all population dynamics, disease and population control processes were spatially organised within territories. The grid was wrapped round to form a torus to eliminate edge effects. The spatial area components were distributed randomly at a realistic density and shape, and new spatial configurations and initial host populations were created for each iteration. Culling-related perturbation of badgers occurred during the initial four-year cull, but, in line with previous findings, was assumed to have stopped by the end of the initial cull and did not therefore occur during any of the follow up strategies.

All events occurred within defined time steps (two month intervals). Most were controlled by parameters that were randomly varied within permitted ranges, allowing stochasticity within the model. The badger population was regulated by parameters including mortality and birth rate. The model permitted disease transmission between individuals within and between social groups to represent respective contact rates. The simulated prevalence in the badger population stabilised at 14% prior to badger control (within the range normally seen in the wild [13]). It was not necessary to simulate badger social perturbation during the follow-up control strategies as the effects of perturbation had ceased before the end of the initial four-year cull.

## Badger management

The model ran for a number of years until stabilised and all data were then stored so that a number of different management options could be run from an identical starting point. A central area of ~400km$^2$ was chosen for badger control. The proportion of compliant accessible land within the intervention area was set to 70% to maintain consistency with the minimum license requirements [14, 15]. Annual trapping efficacy was set at 70% to simulate field estimates for this method [16]. Badger social groups that were situated within or partly within participating farms in the intervention area were identified and subjected to intervention.

Intensive badger culling was simulated for four years (2016–2019), followed by alternate badger culling or vaccination which continued indefinitely. In the alternate culling scenarios, animals were removed from the population at random at a probability determined by trapping efficacy. This occurred (1) every year to simulate the current supplementary culling, (2) every second year, (3) every third year, or (4) two years followed two years without. Trapping efficacy was adjusted for each of these strategies to maintain a low badger population: for annual supplementary culling this was approximately 27%. This was compared to a strategy which simply stopped all badger culling after the four year intensive cull. For the vaccination scenario, animals were trapped with the same probability, vaccinated and released once per year. Each time a healthy badger was vaccinated, it was given a 70% probability of becoming fully and permanently protected against bTB [5]. It was assumed that vaccination had no effect on an animal that was already infected with bTB. Additionally we assumed a strategy was possible in which all animals trapped annually for vaccination were also given a fertility control agent which stopped them from breeding for life [e.g. 17].

## Data output

The model was run with 100 simulations from the same starting conditions. Results from each intervention, and a scenario where no further action after the initial four-year cull was taken were recorded. Key output parameters were mean badger social group size, mean number of infected badgers per social group, mean prevalence and numbers of animals removed or vaccinated.

## Results

The simulated population was recorded immediately prior to control, as though recorded by badger trapping teams. Output is shown for the central controlled area from 2015–2050, where intensive control commenced in 2016 to 2019, followed by the various exit strategies which then continued indefinitely. During the intensive cull the mean number of badgers removed in each year was 1648 (2016), 831 (2017), 525 (2018) and 390 (2019). Thus, the mean badger social group size declined rapidly during the initial cull, but thereafter it recovered to its starting size after about ten years of vaccination or no culling (Fig 1). The intermittent culling options maintained the mean social group size at a low, but variable amount, as did vaccination with fertility control. The exact level of culling in these strategies could not be easily matched to ensure they were identical, but this had no real effect on the amount of disease in the population.

Since the total population size differed markedly between these exit strategies, we present the mean number of infected badgers, rather than prevalence. Following an identical reduction in the number of infected badgers during the initial cull, only the option of no further management led to a marked increase in the number of infected animals (Fig 2), but this increased at a rate much slower than the population recovery. All the remaining strategies led to a further slow decline in the number of infected badgers, and little apparent difference between them.

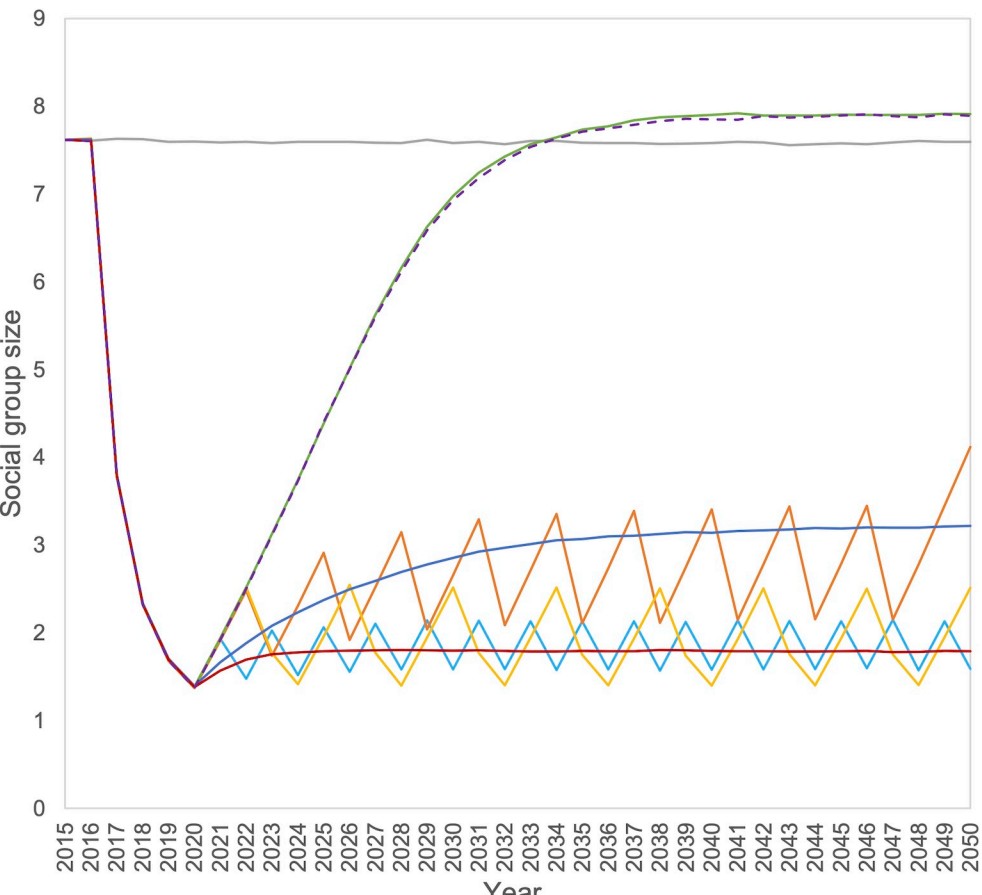

**Fig 1.** Mean badger social group size for the seven scenarios; no badger control (grey line), and intensive cull followed by no control (purple dashed line), annual supplementary culling (dark blue line), biannual culling (light blue line), culling every third year (orange line), culling for two out of four years (yellow line), vaccination (green line) and vaccination with fertility control (red line).

Each of the culling strategies removed 225 badgers per year on average, with no real difference between them (range 207–247 per year). However, the number of badgers trapped and vaccinated increased from ~300 to about 1700 per year after ten years. If combined with fertility control then the numbers trapped and vaccinated increased to a maximum of about 400 per year, due to the very limited birth rate.

## Discussion

In this study we used an established model to simulate bTB dynamics in badger populations in an area of high badger density (the High Risk Area of England) after the end of a four-year intensive badger culling policy. The simulated intensive control removed 1648 badgers in year 1, decreasing to 525 in year 3, which is very similar to the mean numbers removed in the English badger control operations [18–21], so we can be confident that the simulation is accurate for the intensive control period. A number of potential badger management interventions were simulated, including different ongoing or intermittent badger culling to maintain a low population, vaccination, and vaccination with fertility control [an option not yet available in the field, but under investigation, 17]. We assumed similar capture efficiency (i.e. 70%) and access to land (70%) for all strategies to ensure they were directly comparable.

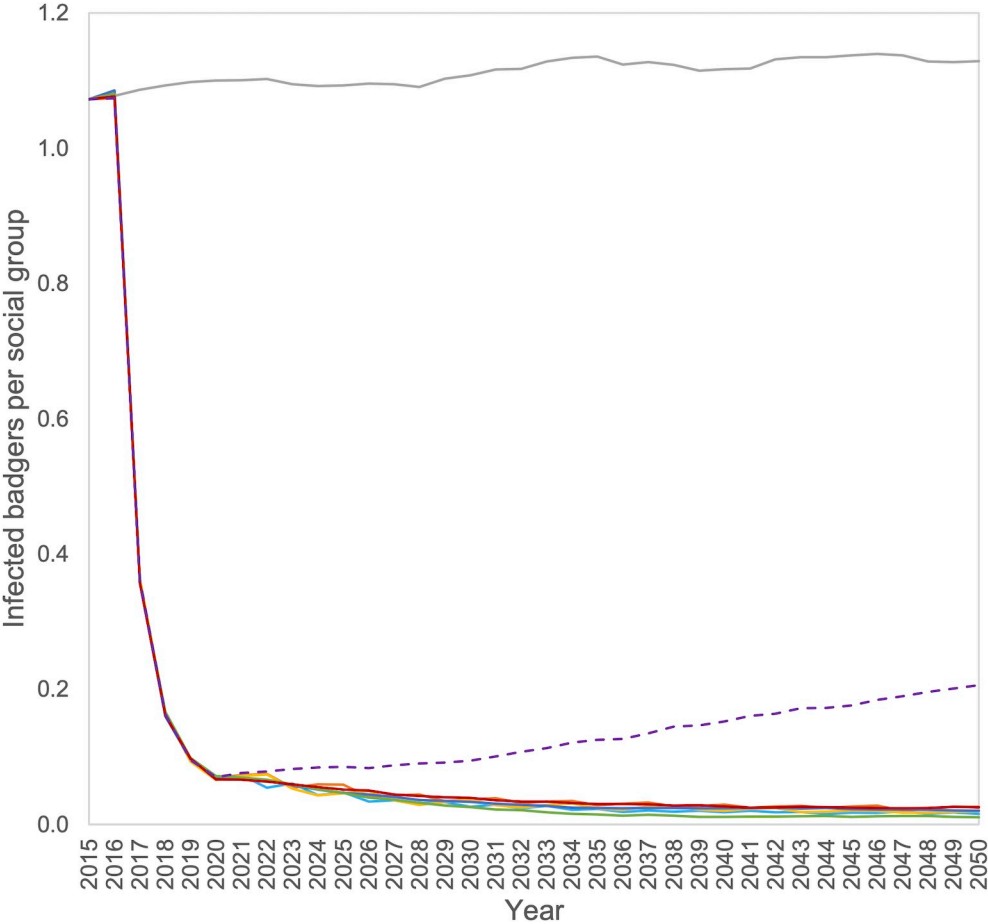

**Fig 2.** Mean number of infected badgers per social group for the seven scenarios; no badger control (grey line), and intensive cull followed by no control (purple dashed line), annual supplementary culling (dark blue line), biannual culling (light blue line), culling every third year (orange line), culling for two out of four years (yellow line), vaccination (green line) and vaccination with fertility control (red line).

Although there is evidence of a positive effect of vaccination on TB progression in badgers [6] there is no empirical data on how vaccination of badgers impacts on cattle bTB incidence, so in these scenarios we only examined the effect of badger management on bTB infection in the badger population and by extrapolation, assume that the absolute number of infected badgers is a monotonic index of infection risk in cattle.

The initial intensive cull reduced the number of infected badgers within the central controlled area to about 10% and population size to about 20% of pre-control levels. We have no way to determine the accuracy of this since population size and prevalence have not been determined in any of the current control areas. However, the exact level of reduction is not important, since this occurred prior to the exit strategies simulated here. Badger bTB prevalence was reduced to about one third of its original level by the end of the intensive cull, and continued to decrease with all exit strategies, but was dependent on population size hence the lowest prevalence was seen with vaccination only (scenario 5) as the population size increased under this strategy. Hence we reported the number of infected badgers, as this is a more accurate reflection of risk to cattle, rather than population size or prevalence. However prevalence can be estimated from a sample of badgers and as such is the measure of infection most likely to be available in the field, so in practice some estimate of population size would be needed.

All control scenarios maintained a reducing level of infection in badgers for the duration of exit strategy control, implying bTB eradication may be possible in the long term. All scenarios involving population suppression maintained the badger population at a stable level around 30% of the pre-culling level. It is possible the badger population could be maintained in the field at around this target with further modification of the removal rate or periodicity of culling, and with knowledge of exact number of animals removed and an estimate of the size of the remaining population.

The low infection levels predicted are likely to be the result of the high intensity of the initial and ongoing removal operations, and the low rate of reinvasion, even with neighbouring infected badger populations. We assumed permanent movement of badgers was limited to neighbouring territories, as these were below carrying capacity so reinvasion across the area culled was slow. An additional benefit to consider is that many of these badger control areas in England now abut with each other, further reducing immigration. If there is spatial heterogeneity in future management strategies, then the effect of immigration may become more important where an increasing population (under a no-cull or vaccination approach) abuts to an ongoing culled population.

The badger vaccination strategies offer a welfare advantage by avoiding culling. None of the culling strategies offered a welfare advantage by culling fewer animals, as the same total number was removed in each culling strategy, however cost and efficiency may be different between scenarios depending upon the number of repeat visits, time taken at each trap visit, etc.

The largest number of animals were removed during the four-year intensive culling period and subsequent removal operations were smaller.

The total number of animals trapped for vaccination was much higher than for culling as the population recovered over time. When vaccination was given with fertility control, the total number of badgers trapped was less than half the number trapped using vaccination on its own.

All BCG vaccination simulations assumed complete individual protection for a portion of vaccinated animals. The current evidence suggests that while this is true for some, others may still become infected but show reduced progression. This may reduce the overall efficacy of vaccination compared to culling and to the model is being adjusted based on the latest data to examine this in more detail.

Overall, the model output during the intensive cull is very similar to the field data, suggesting our starting point for future badger management is accurate. If no badger management is performed, then the population recovers in about 10 years but the absolute number of infected badgers takes much longer to increase. All management approaches could maintain the badger density at less than half of its carrying capacity, except for vaccination, where the population also recovered in about 10 years. All management strategies were capable of reducing the absolute number of infected badgers over time, with little to choose between them in this metric. This includes vaccination during which the population size increases. Therefore the main difference is that vaccination does not involve killing any badgers, although the number of badger to be vaccinated will increase over time. The cost in the short term would be similar for all the above strategies, but note that the increase in population size for a vaccinated population would slowly increase the numbers trapped that require a vaccine, which would slightly increase the cost. In the longer term vaccination combined with fertility control may also be a useful management tool to retain a reduced population size.

## Supporting information

**S1 Appendix. Model variables (temporal settings).**
(DOC)

**S2 Appendix. Model variables (spatial settings).**
(DOC)

**S3 Appendix. Model variables (badger parameters).**
(DOC)

**S4 Appendix. Model processes (submodels).**
(DOC)

## Author Contributions

**Conceptualization:** Graham C. Smith.

**Data curation:** Richard Budgey.

**Formal analysis:** Richard Budgey.

**Funding acquisition:** Graham C. Smith.

**Investigation:** Richard Budgey.

**Methodology:** Graham C. Smith.

**Project administration:** Graham C. Smith.

**Software:** Richard Budgey.

**Supervision:** Graham C. Smith.

**Validation:** Richard Budgey.

**Visualization:** Richard Budgey.

**Writing – original draft:** Graham C. Smith.

**Writing – review & editing:** Richard Budgey.

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
