## [Decision Letter · Decision Letter 0]

7 Oct 2020

PONE-D-20-26898

Simulating the next steps in badger control for bovine tuberculosis in England

PLOS ONE

Dear Dr. Smith,

Thank you for submitting your manuscript to PLOS ONE. After careful consideration, we feel that it has merit but does not fully meet PLOS ONE’s publication criteria as it currently stands. Therefore, we invite you to submit a revised version of the manuscript that addresses the points raised during the review process.

Please submit your revised manuscript. If you will need more time than this to complete your revisions, please reply to this message or contact the journal office at plosone@plos.org. Please include the following items when submitting your revised manuscript:

We look forward to receiving your revised manuscript.

Kind regards,

Frederick Quinn

Academic Editor

PLOS ONE

Journal Requirements:

Reviewers' comments:

Reviewer's Responses to Questions

**Comments to the Author**

1. Is the manuscript technically sound, and do the data support the conclusions?

Reviewer #1: Yes

Reviewer #2: Yes

2. Has the statistical analysis been performed appropriately and rigorously? 

Reviewer #1: I Don't Know

Reviewer #2: N/A

3. Have the authors made all data underlying the findings in their manuscript fully available?

Reviewer #1: Yes

Reviewer #2: Yes

4. Is the manuscript presented in an intelligible fashion and written in standard English?

Reviewer #1: Yes

Reviewer #2: No

5. Review Comments to the Author

Reviewer #1: This useful and timely analysis compares different models of controlling badger populations, to reduce risk of BTB spread to cattle. The main claims of the paper are that the 3 control approaches are projected to result in similar reduction in infected badgers, based on statistical modeling.

The data and analyses support the claims.

However, the manuscript could be revised to make it clear to non-specialists in the field.

For example, on page 2 authors could elaborate further on reference 10 about the desirability of non-lethal control models. BCG vaccination protocols could be briefly explained as 2 options (injection versus oral baiting) could be utilized.

The Discussion ends without a summary of conclusions, which are found only in the Abstract. Assuming the 3 control methods are equivalent, then the authors should advance how wildlife managers would make a selection, i.e. cost analyses as well.

Reviewer #2: The modeling work presented in the paper addresses the important question of the benefit of deploying vaccines to control TB in badgers and cattle in the UK, under the current situation of industry-deployed cull.

The vaccine considered for the model is injectable (by IM delivery), which should be mentioned (for ex on L103 and L157) since it implies the need for trapping badgers, contrarily to oral vaccination which could be more widely deployed and potentially reach a larger range of animals/groups.

The model is based on the TB situation in the UK, which should be specified more in the paper, for exemple l41 “it is not certain what level of effect this would have on bTB incidence in cattle in the UK”.

The output of number of infected badgers only is surprising, especially as the author mentions in L173-174 that prevalence is the measure most likely to be available in the field. The author may be more explicit on the rational for choosing this output, possibly that the risk to cattle may be expected to be proportional to the number of infected animals, not to their density, and this should be discussed.

Finally, the overall conclusion could be clearer. Is vaccination a favorable strategy compared with culling?

Corrections and new formulations are suggested below:

L14: Justify why the prevalence of bTB in badger population was fixed at 14%

L108, is the fertility control treatment vaccination given yearly in the animals trapped for injectable vaccination, as suggested in the legend of fig1?

Rephrase L173-174.

L170: replace reduce by decrease

L175: add “in badgers” after “infection”

L176: add “bTB” before eradication

Rephrase L179-180.

Rephrase L184 from “so reinvasion was slow”.

L184, be more specific about the strategy with additional benefit. The author should also mention the lack of immigration in the context of vaccination, given the return of the population number to social equilibrium between the groups.

L187 The author mentions the (non existant) welfare advantage of culling strategies but not of vaccination exit strategy involving vaccination .

L192, add” than for culling” after “was much higher”

L193, Replace “whereas” by “when”, rephrase after “this resulted”

L195, add “with BCG” after vaccination and “individual” before protection, and “vaccinated” before animals

L198, be more precise in term of other work in hand to examine the overall efficacy of vaccination

6. PLOS authors have the option to publish the peer review history of their article (what does this mean?). If published, this will include your full peer review and any attached files.

Reviewer #1: No

Reviewer #2: No

---

## [Author Response · Author response to Decision Letter 0]

8 Feb 2021

All reviewers comments have been adressed

---

## [Decision Letter · Decision Letter 1]

26 Feb 2021

Simulating the next steps in badger control for bovine tuberculosis in England

PONE-D-20-26898R1

Dear Dr. Smith,

We’re pleased to inform you that your manuscript has been judged scientifically suitable for publication and will be formally accepted for publication once it meets all outstanding technical requirements.

Kind regards,

Frederick Quinn

Academic Editor

PLOS ONE

Additional Editor Comments (optional):

Reviewers' comments:

Reviewer's Responses to Questions

**Comments to the Author**

1. If the authors have adequately addressed your comments raised in a previous round of review and you feel that this manuscript is now acceptable for publication, you may indicate that here to bypass the “Comments to the Author” section, enter your conflict of interest statement in the “Confidential to Editor” section, and submit your "Accept" recommendation.

Reviewer #1: All comments have been addressed

2. Is the manuscript technically sound, and do the data support the conclusions?

Reviewer #1: Yes

3. Has the statistical analysis been performed appropriately and rigorously? 

Reviewer #1: I Don't Know

4. Have the authors made all data underlying the findings in their manuscript fully available?

Reviewer #1: Yes

5. Is the manuscript presented in an intelligible fashion and written in standard English?

Reviewer #1: Yes

6. Review Comments to the Author

Reviewer #1: (No Response)

7. PLOS authors have the option to publish the peer review history of their article (what does this mean?). If published, this will include your full peer review and any attached files.

Reviewer #1: No

---

## [Editor Report · Acceptance letter]

9 Mar 2021

PONE-D-20-26898R1 

Simulating the next steps in badger control for bovine tuberculosis in England

Dear Dr. Smith:

I'm pleased to inform you that your manuscript has been deemed suitable for publication in PLOS ONE. Congratulations! Your manuscript is now with our production department. 

Kind regards, 

on behalf of

Dr. Frederick Quinn 

Academic Editor

PLOS ONE